# Characteristic Analysis of Four Major Nighttime Fire Cases on Fire Safety of Long-Term Care Institutions Using Fire Protection Defense-In-Depth Strategy

**Liang-Sheng Wu [1], Ryh-Nan Pan [2]** **, Shiuan-Cheng Wang [3], Chung-Hwei Su [1,\*] and Wen-Chien Wu [1]**

1 Department of Safety, Health and Environmental Engineering, National Kaohsiung University of Science and Technology, Kaohsiung City 824005, Taiwan

2 Department of Fire Safety, Taiwan Police College, Taipei City 11696, Taiwan

3 Department of Public Safety and Fire Science, Chia Nan University of Pharmacy and Science, Tainan City 717301, Taiwan

\* Correspondence: such@nkust.edu.tw; Tel.: +886-7-601-1000 (ext. 37613)

**Abstract:** Providing high-quality care services and fire safety for long-term care institutions is an important issue in Taiwan, which became an aging society in 2018. The fire incidents in Taiwan over the years show that nighttime fires in care institutions often cause serious casualties. It is necessary not only to understand the causes of serious nighttime fire incidents that have occurred but also to draw lessons from the fires that have been put out without causing injuries. In this study, the top two serious nighttime fire accidents in long-term care institutions in the past two decades in Taiwan were analyzed based on the publicly official and academic literature utilizing fire protection defense-in-depth strategies. For comparison, two other nighttime fire cases with similar scenarios but no casualties were also analyzed in depth about the cause of no casualties. The buildings of the four nighttime fires were equipped with fire protection equipment in their public areas. The theoretical basis of the research is the fire protection defense-in-depth strategy. In both categories of severe casualties and no severe casualties, one was caused by arson and the other one by an electrical fire, with the ignition point of a fire in the storeroom and the other in the ward. However, the end results were quite different. The analyzed results showed that the severe fires lasted for about an hour, while the fires without casualties were put out within 15 min. A well-constructed second layer of defense measures could effectively contain a fire, and an effective third layer of measures could avoid casualties. The death rate of personnel can be reduced from a dozen to zero, and the burning time is also greatly reduced. The results could be used as a reference for emergency measures in long-term care institutions.

**Keywords:** long-term care institution; nighttime fire incident; fire safety; fire protection defense-in-depth strategy; electrical fire; arson



## 1. Research Motivation

### 1.1. Nighttime Fire Safety Concerns in Long-Term Care Institutions

Taiwan has entered an aging society, with the elderly population accounting for 14.5% in 2018 [1,2]. According to the Ministry of Health and Welfare, 794,050 people needed long-term care in 2019 [3]. The results of Chien and Shi's research show that nighttime fires in Taiwan's care institutions often cause serious casualties. From 2011 to 2019, there were a total of 47 fires in Taiwan's hospitals, nursing homes, senior citizens' welfare institutions, veterans' homes, and welfare service centers for the disabled, resulting in 44 deaths and 182 minor and major injuries [4]. Among the cases, 42% of the casualties (43 deaths and 170 injuries) occurred during the night shift, and 5.8% (one death and 12 injuries) occurred during the night shift. On the other hand, there were no casualties during the daytime.

Strengthening fire safety in institutions is an important issue for all countries. In 2016, Haeri et al. conducted a survey of fire supervision and awareness of fire regulations among 350 staff members in 30 nursing homes in South Korea. The results show that the fire will cause serious damage if the staff's evacuation safety awareness is low. [5]. In 2017, Kim used simulation software to conduct a safety assessment of the Y Welfare Center for the Elderly in Busan and made some suggestions for improving the safety management of the Welfare Center for the Elderly. [6]. In 2020, Chien et al. pointed out that medical institutions and long-term care institutions cannot maintain fire safety in accordance with general fire regulations [7]. As shown in these studies, nighttime fire safety concerns in institutions are summarized into the following categories:

a. People in deep sleep are slow to sense incidents and the light is dim, so their abilities to react to incidents are weak.

b. Most occupants in institutions suffer from various chronic diseases, such as hypertension (HT), coronary artery disease (CAD), diabetes mellitus (DM), cerebrovascular accidents (CVA), chronic kidney disease (CKD), and chronic heart failure (CHF), and need to take medicine such as tranquilizers or sleeping pills regularly [8,9]. These make residents move slowly and are unfavorable for evacuation.

c. In addition to having fewer staff at night, staff on duty are mostly females with smaller strength, which makes them unable to effectively assist in evacuating occupants in the case of an emergency situation.

Arson and electrical fires are two main causes of fires in care institutions and should be put into consideration when drawing up disaster prevention and relief measures. In 2016, Wu and Tseng pointed out that developing effective and performable response strategies is an issue for Taiwan and other countries entering an aging society [10]. Nardo et al. presented qualitative and quantitative fire and explosion risk assessments from the use of liquefied petroleum gas cylinders in domestic environments [11].

### 1.2. Establishment of Disaster Resilience

Taiwan is exposed to more than three categories of natural disasters according to the report of the World Bank [12]. Life-support and care services in Taiwan's long-term care institutions may be interrupted by typhoons, earthquakes, fires, or compound disasters. Occupants who cannot take refuge by themselves, such as patients with chronic diseases and elderly people, face serious threats regarding their medical care quality and life safety in these situations [13,14].

In the fire incident in Brazil's Rio De Janeiro Badim Hospital on September 12, 2019, some critically ill patients could be moved only after their life-support equipment was removed. Some patients were killed by smoke, and some died in the fire because their life-support equipment stopped working [15]. This tragedy showed that that medical institutions and care institutions should continue to pay attention to care quality and occupants' safety needs in the face of disasters.

Resilience refers to the ability to predict, plan, and reduce disaster risks, in order to effectively protect individuals and groups, and it involves culture, society, the economy, and the ecosystem. In the context of disaster risk, resilience refers to the ability of an individual or group exposed to hazards to resist, absorb, accommodate, transform, and recover from the effects of a hazard in a timely and efficient manner. The research results by O'Brien et al. and Huq et al. found that reducing vulnerability is a key aspect of reducing climate change risk [16,17]. In 2015, the Sendai Framework for Disaster Risk Reduction 2015–2030 was proposed at the Third United Nations World Conference on Disaster Risk Reduction, indicating that it is necessary to promote disaster risk identification and build resilient societies [18]. The primary goal is to reduce disaster-related mortality and the number of people affected by disasters, and the second goal is to reduce the direct economic losses caused by disasters.

As Taiwan's care institutions are at risk of nighttime fires, a review on domestic disaster cases is necessary. This study investigated four representative cases which are the

worst and most likely to happen and conducted hazard identification. The analysis results could help institutions make improvements according to their conditions and establish an overall safety plan.

### 1.3. Disadvantaged Groups in Evacuation during Nighttime Fires

Taiwanese scholars Yi-Yung Yang and Ching-Yuan Lin defined disadvantaged groups in evacuation as people who are mentally impaired, physically impaired, or move slowly, with abilities to react to and egress from disasters that are lower than ordinary people. These people have low abilities to recognize and act in their environments in ordinary times [19].

Disadvantaged groups in evacuation during nighttime fires need to be handed or use assistive devices (such as wheelchairs), stretchers, and beds to move smoothly. Due to their poor mobility, the chances of survival are lower than those of common people in case of an emergency. Chen collected the characteristics of cognition and mobility of disadvantaged groups in evacuations, as shown in Table 1 [20].

**Table 1.** Characteristics, cognition, and mobility of various disadvantaged groups during evacuations.

| Category of Disability | | Disability Characteristics | Cognition |
|---|---|---|---|
| Mental retardation | | Insufficient ability to identify and recognize information; slow moving function and reaction | Information recognition |
| Visual impairment | | Inability to recognize shapes of objects; narrow field of vision; abnormal light perception; inability to distinguish colors | Information recognition impairment Mobility impairment |
| Hearing impairment | | Hearing impairment; poor sensitivity to sound; inability to receive sound information and audible signals | Information recognition |
| Limb deficiency | | Limb or trunk deformity; inability to move joints or to stand; the necessity to use assistive devices such as crutches and wheelchairs | Information recognition impairment Mobility impairment |
| Multiple impairments | Visual and hearing impairments | Having both visual and hearing impairments | Information recognition impairment Mobility impairment |
| | Cerebral palsy (mental and limb deficiencies) | Inability to identify information, insufficient cognitive ability, slow moving function and reaction, and limb deficiencies | Information recognition impairment Mobility impairment |

## 2. Research Method

### 2.1. Fire Protection Defense-in-Depth Strategy

Nuclear power plants are energy-efficient buildings but expensive to construct and maintain. The release of radioactive materials (such as Cesium-137 and Iodine-131) causes serious damage in the event of accidents. The Chernobyl nuclear disaster in 1986 and the Fukushima-Daiichi Nuclear Power Station accident caused by the tsunami in 2011 are examples [21,22]. After the Fukushima-Daiichi Nuclear Power Station accident on March 11, 2011, the Japanese government spent 11 years on decontamination, but it is still unknown when it will fully recover [23].

In the United States, nuclear materials for nonmilitary use are regulated by the U.S. Nuclear Regulatory Commission. In order to avoid serious incidents, the fire protection defense-in-depth strategy has been developed to protect people from the danger of nuclear material leakage. The mechanism provides layers of protection. If one layer fails, there

will be another layer of protection [24,25]. The strategy considers that any complex close-coupled system, no matter how well-engineered, cannot be said to be failure-proof. In 2011, Saleh and Cummings used the defense-in-depth theory to discuss the safety improvement of the mining industries [26]. In 2020, Papakonstantinou et al. utilized the theory of defense-in-depth as a safety and security assessment approach to eliminate loopholes and weaknesses in defense [27].

The failure of layers of protection may be due to negligence or equipment malfunction, and negligence may happen at any time a nuclear plant is in operation [28]. Figure 1 shows the concept of three layers of protection:

Layer 1: To prevent fires from starting.
Layer 2: To rapidly detect, control, and extinguish fires that do occur.
Layer 3: To protect the nuclear reactor so that fires which are not promptly extinguished by the fire suppression activities do not prevent the safe shutdown of the plant. With regard to the third layer, this study focused on preventing the extension of disasters.

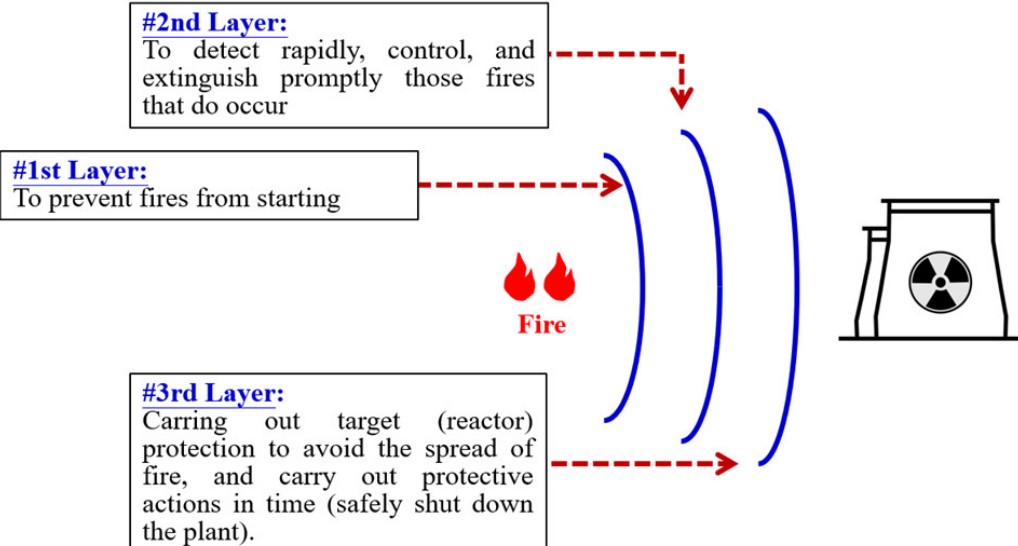

**Figure 1.** Fire protection defense-in-depth strategy in U.S. nuclear power plants.

*2.2. Timed Egress Analysis Method*

The safety of people's evacuation must be taken seriously according to the Building Act in Taiwan; as a result, entrance and exit widths, corridor widths, and stair widths are all regulated. The timed egress analysis method, widely used internationally, evaluates people's safety using factors such as the number of people, people's horizontal or vertical movement velocity, and the length of time when the event reaches a dangerous level. The explanation is shown in Figure 2 [29].

The analysis on people's evacuation behavior in fires can be divided into three steps: (1) detection of signs of fire; (2) confirmation of fire; and (3) evacuation behavior. In 2020, Ku and Chow described that fire evacuations are usually designed using timeline analysis. Available safe egress times (ASET) and required safe egress times (RSET) can be compared with agreed scenarios.

Institutions where fires occurred have set up hardware equipment and formulated management measures in accordance with laws and regulations. Nursing staff have also undergone regular training in accordance with regulations. In 2019, Chiu et al. conducted an analysis of the factors affecting the fire strain in long-term care institutions and obtained a lot of data. The ranking of the obtained evaluation indicators is the differentiated drills in various situations, self-defense fire formation and initial contingency drills, and other indicators [30]. According to statistics from Taiwan's Fire Department in 2021, the

ambulance response time will be 56.39% within 6 min; 76.74% within 8 min; and 87.60% within 10 min [31].

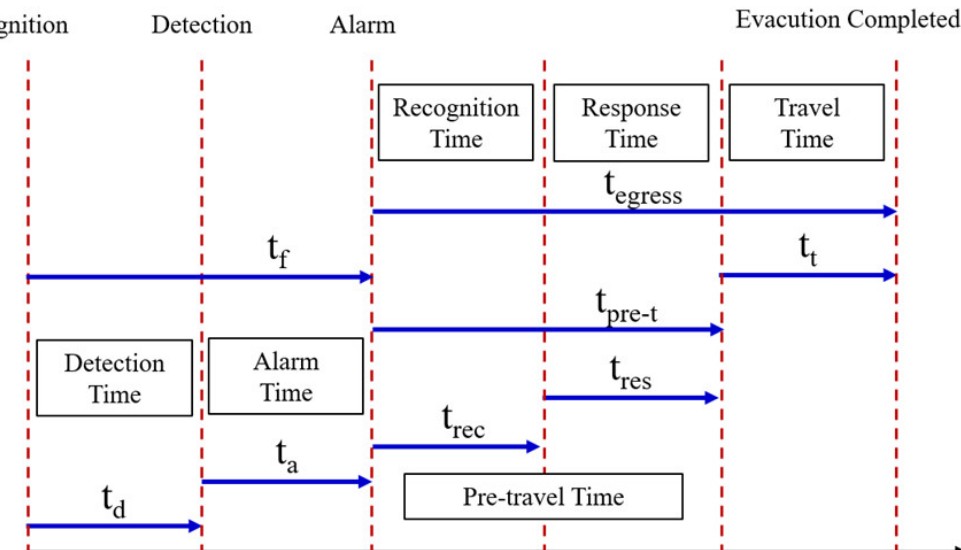

**Figure 2.** Flow chart of timed egress analysis method.

The relationship between the available safe egress time (ASET) and the required safe egress time (RSET) were compared in this study to determine whether people can egress from fires [32,33]:

a.    *Available safe egress time (ASET)*

ASET represents the period th from the time a fire starts to the time the fire poses risks to people, as shown in Equation (1):

$$\text{ASET} = t_h \tag{1}$$

The thermal radiation, thermal energy, smoke, and toxic gases in fires are harmful to human bodies. In fire protection engineering, quantified physical parameters such as thermal radiation flux ($kW/m^2$ or $W/cm^2$), air temperature (°C), the concentration of CO or other toxic gases (%), smoke layer height (m), and visibility (m) are measured to show the damage degree.

b.    *Required safe egress time (RSET)*

RSET represents the time from the start of a fire and the evacuation of people to safe areas, including the fire detection time ($t_f$), pre-evacuation time ($t_{pre-t}$), and evacuation time ($t_t$), in which t$_f$ includes the detector induction time ($t_d$) and alarm time ($t_a$). The pre-evacuation time ($t_{pre-t}$) can be divided into the fire recognition time ($t_{rec}$) and people response time ($t_{res}$). The evacuation time ($t_t$) is related to characteristics such as people's movement velocity (V, m/s), people's distribution density (D, person/$m^2$), and people's mobility (Q, person/(m·s)), as shown in Equation (2) [34,35]:

$$\text{RSET} = t_f + t_{pre-t} + t_t = t_f + (t_{rec} + t_{res}) + t_t \tag{2}$$

Equations (1) and (2) can be compared and expressed as follows:

$$\text{RSET} \leq \text{ASET} \tag{3}$$

$$\frac{t_f + (t_{rec} + t_{res}) + t_t}{t_h} \leq 1 \tag{4}$$

According to the fire protection defense-in-depth strategy, the second and third layers of protection should be strengthened in the case of fire disasters. Increasing the safety margin time can improve fire safety in care institutions. Good building of fire protection facilities and equipment can be prepared to provide the function of safe refuge, including [36]:

1.  Prolonging the risk factor ($t_h$): the initial firefighting and well-planned fire compartmentation should be carried out immediately to avoid fire expansion and prolong the time before the evacuation environment is damaged. These two items are the second and third layers of protection, respectively. If the hazard time ($t_h$) increases or even becomes infinite ($\infty$), occupants in the institution will have more time to evacuate.

2.  Reducing control factors ($t_f$, $t_{pre-t}$, and $t_t$): $t_f$ is the second layer of protection, which is based on informing people of a fire as soon as possible. $t_{pre-t}$ is also the second layer of protection. The time for fire confirmation can be reduced by broadcasting evacuation instructions immediately. Item 3 ($t_t$) refers to a well-planned evacuation path and appropriately established evacuation guidelines, which can reduce people's movement time. However, this is related to the characteristics of the people. In institutions, occupants with mobility impairments can only be evacuated with assistance. It is very difficult to reduce $t_t$ at night when there are limited personnel in the institution. According to the review of case data and interviews with staff at the emergency scene, the goal of lowering $t_t$ is challenging. This study explored the protection of occupants who cannot move using the fire protection defense-in-depth strategy. Therefore, it is reasonable to assume that attention to care recipients is as important in a fire emergency as a nuclear reactor when a fire occurs in a nuclear power plant.

## 3. Disaster Description for Major Nighttime Fire Cases

In the past 20 years, there have been nighttime fire cases involving casualties in hospitals and long-term care institutions according to statistics from the Taiwan Fire Department. The fire incidents in Taiwan over the years show that nighttime fires in care institutions often cause serious casualties, since most occupants in these institutions are mobility-impaired and have fallen asleep and few staff members are available to assist in the evacuation. Arson is the most adverse but possible situation in institutions. The second is fire caused by electrical appliances [4,37,38].

Some researchers of this study have worked in fire control units, and some have served as members of the fire cause investigation committee and have carried out investigations at fire scenes, including cases without casualties. Based on publicly official literature and academic theories, the top two serious nighttime fire accidents in long-term care institutions in the past two decades in Taiwan were analyzed. For comparison, two other nighttime fire cases with similar scenarios but no casualties were also analyzed in depth about the cause of no casualties. The theoretical basis of the research is the fire protection defense-in-depth strategy. The buildings of the four nighttime fires were equipped with fire protection equipment in their public areas according to fire law. This study explained the common characteristics of four cases as follows. The comparable information of the four cases is shown in Table 2.

*   Among the four cases, there were serious casualties in two representative cases in Taiwan (Cases 1 and 3) and no casualties in the other two cases (Cases 2 and 4).
*   The fires occurred in the storerooms in two cases (Cases 1 and 4) and in the wards in the other two cases (Cases 2 and 3). However, the results were different.
*   Arson was the cause of fire in two cases (Cases 1 and 2) and electrical fires occurred in the other two cases (Cases 3 and 4). However, the results were different.

**Table 2.** The number of websites where the incidents were mentioned.

| Case No. | Name of Nighttime Fire Accident | Statistics Period | | Result Value of Web Query (Search with Google) |
| | | Started | Ended | |
|---|---|---|---|---|
| Case 1 | Xinying Hospital Beimen Branch fire | 23 October 2012 | | 76 |
| Case 2 | Tainan Hospital Psychiatric Rehabilitation Ward fire | 10 May 2015 | 30 September 2022 | 18 |
| Case 3 | Taipei Hospital nursing home fire | 13 August 2018 | | 66 |
| Case 4 | Kaohsiung St. Joseph Hospital nursing home fire | 20 April 2022 | | 29 |

### 3.1. Case 1: 2012 Tainan Xinying Hospital Beimen Branch Fire

This case took place in the early hours of 23 October 2012. A fire broke out at MOHW Xinying Hospital in Tainan City's Beimen District, killing 13 people and injuring 61 others. The fire raised social concerns, as the investigation indicated it was an arson event. The fire started in a maternity ward. Due to poor business, it had been idle for 14 years and had become a storeroom. According to Taiwan's fire code, storerooms do not need to be equipped with automatic sprinkler systems. This case was a serious casualty incident in a public hospital, and the hospital officials and the Ministry of Health and Welfare responsible for supervision were required to conduct reviews and make improvements [39]. Some information is shown in Table 3, including the fire location, cause, burning time, rescue personnel, and the number of casualties.

**Table 3.** Comparison and explanation of four nighttime fire cases.

| Item | Serious Casualty Cases (Case 1, 3) | | No Casualty Cases (Case 2, 4) | |
| | Xinying Hospital Beimen Branch | Taipei Hospital Nursing Home | Tainan Hospital Psychiatric Rehabilitation Ward | Kaohsiung Shenggong Hospital Nursing Home |
|---|---|---|---|---|
| Date | 03:29, 23 October 2012 | 04:29, 13 August 2018 | 00:58, 10 May 2015 | 05:44, 20 April 2022 |
| Institution type | Nursing Home | Nursing Home | Psychiatry rehabilitation institution | Nursing Home |
| Fire floor | 2F | 7F | 1F | 3F |
| Location and cause | A storage room on the 2nd floor (originally used as a delivery room), the patient set fire to the quilt | Electric appliance burns in ward on the 7th floor | Patient sets fire in mental ward on 1st floor | The quilt was burnt due to an electrical fire in the storage room of the tracheostomy wards area on the 3rd floor |
| Fire Safety Equipment | Fire extinguishers, fire detectors, automatic sprinkler equipment, smoke exhaust equipment | | | |
| Rescue vehicles and personnel | 53 fire trucks, 48 ambulances (including 26 support ambulances from neighboring cities), and 150 firefighters | 76 vehicles, 236 firefighters | 10 vehicles, 26 firefighters | 19 vehicles, 43 firefighters |
| Fire duration | About one hour | About one hour | About 10 min | About 15 min |
| Casualties | 13 killed, 61 injured | 15 killed, 37 injured | No casualties | No casualties |
| The state of room door | The door of the fire room was not fully closed | | The door to the fire room is completely closed | |
| Failure level in Fire Protection Defense-in-Depth | Level 1, 2, 3 | Level 1, 2, 3 | Level 1, 2 | Level 1 |

### 3.2. Case 2: 2015 Tainan Hospital Psychiatric Rehabilitation Ward Fire

A fire broke out at the psychiatry ward in MOHW Tainan Hospital in the early hours of 23 October 2018. The investigation revealed the fire started in a psychiatry ward on the first floor. A total of 48 patients were evacuated to the plaza. The fire was put out within ten minutes, with no casualties. Only one bed was burned. A patient lit a book with a lighter, igniting his bed sheet and causing a fire. An occupant in his ward was unable to detect the fire immediately because he had taken sleeping pills.

Because the ward was equipped with a fire detector, it sensed the fire and gave an alarm. The fire was too big for nursing staff to put out with fire extinguishers because the combustion materials were flammable. The nursing staff closed the fire door and the firefighters arrived later to put out the fire using the hospital's hydrants [40]. Since there were no casualties, hospital officials only recorded the incident and were not required to conduct reviews or make improvements.

### 3.3. Case 3: 2019 Taipei Hospital Nursing Home Fire

A fire broke out at the nursing home of MOHW Taipei Hospital in the early hours of 13 August 2018, killing 15 people and injuring 37. It was a major hospital fire and raised social concerns. All the dead had suffered smoke inhalation during the evacuation. The investigation confirmed that the cause was an electrical fire and not arson, and the electric appliance was the health care device of a patient. The mattress melting at the scene had been used for months and the wires had worn away from long periods of movement, causing a short circuit. The patient had brought the mattress to the ward and used it for months. The management staff were investigated by the judicial authorities for negligence.

The investigation found the management staff had maintained the firefighting equipment in accordance with the fire code, had a disaster prevention plan, and carried out fire drills, so the police decided not to prosecute. However, it was a serious casualty incident in a public hospital, and the Ministry of Health and Welfare (responsible for supervision) was reviewed by the Control Yuan of the Republic of China (Taiwan), which required the Ministry to make improvements [41].

During the investigation of this case, one phenomenon was discussed. All the patients in the room where the fire started had suffered smoke inhalation but there were no casualties, while all the dead were from other rooms. The analysis revealed the cause to be hot smoke that spread in the space above the ceiling, as shown in Figure 3. The building code implemented when the hospital was built allowed the walls to be connected only to the ceiling but not to the floor above. This phenomenon was the same as that in Case 1.

### 3.4. Case 4: 2022 Kaohsiung St. Joseph Hospital Nursing Home Fire

A fire broke out at the nursing home on the third floor of Kaohsiung's St. Joseph Hospital at 5:44 a.m. on a day in 2022. Dispatchers immediately sent 19 vehicles and 43 firefighters to the scene. By the time firefighters arrived, the nursing staff had notified other staff (including doctors on duty, guards, and pharmacy staff) on all floors to move 30 long-term bedridden patients with tracheotomies and mobility impairments to a safe place. There were no casualties. Within 15 min of the alarm sounding, the fire was put out by firefighters [42].

The fire broke out in a storeroom where quilts and bed sheets were kept, after electric equipment caught fire and ignited the stacked items. Further details are being investigated. The fire detector and automatic sprinkler system in the storeroom all worked.

The investigation of the fire scene revealed all walls of the room were connected to the floor above. This situation was the same as that in Case 2. Therefore, the flames and smoke were confined to the room where the fire started and did not spread. However, according to the fire code, only fire detectors were required to be installed in the room in Case 2, and sprinklers were not required.

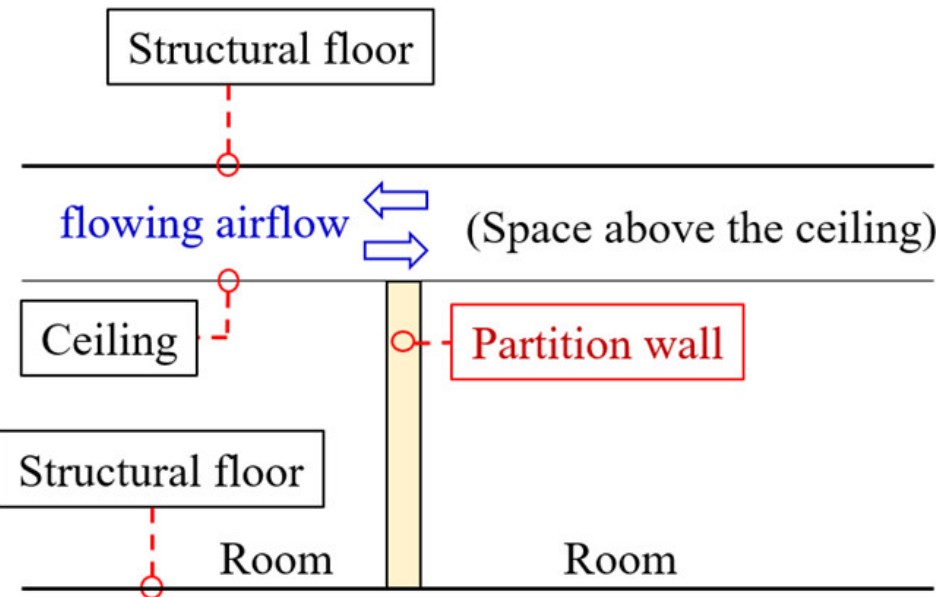

**Figure 3.** The wall is not connected to the floor structure above.

## 4. Results and Discussion

*4.1. Number of Websites about Nighttime Fires*

According to Stat Counter's global statistics, as of April 2022, the top three search engines used in Taiwan were Google, Yahoo!, and Bing, with a market share of 94.98, 3.56, and 1.1%, respectively [43]. As the incidents discussed in this study occurred during the last ten years, this study used Google to collect searchable data about the above four cases on 26 April 2022 as a focus index. The results are shown in Table 4. The people paying attention to these events may include managers and nursing staff of institutions or relatives of occupants.

**Table 4.** Mobility of personnel in a care institution.

| Mobility Group | Description | Average Walking Speed (m/s) | |
| --- | --- | --- | --- |
| | | Horizontally | Stair (Vertically) |
| Persons who are unable to be autonomous or have limited mobility | Bedridden, seriously ill, physically handicapped | 0 to 0.8 | 0 to 0.4 |
| Healthy people unfamiliar with paths inside the building | Accompanying family members | 1.0 | 0.5 |
| Healthy personnel familiar with paths within the building | Doctor, nurse, inspector, security | 1.2 | 0.6 |

The researchers visited the sites of different incidents where there were no casualties and found some advantages worth considering. The cases causing serious casualties were often discussed and written into articles. According to the data, the incidents causing serious casualties (Cases 1 and 3) were obviously of high concern. Data on the incidents causing no casualties (Cases 2 and 4) could only be found on a few websites due to the lack of follow-up discussion.

Case 4 happened recently and was published on many websites, but the number of websites where this case was mentioned was 50% less than the number of websites mentioning serious incidents. This study analyzed the four nighttime fire cases using the fire protection defense-in-depth strategy and made comparisons between the cases with serious results and those with good results, to provide a reference for relevant people and to show admiration for the firefighters in charge of the cases where there were no casualties.

*4.2. Results Analysis and Discussion*

All the four fires broke out between midnight and 6:00 in the early hours, while people were asleep. Institutions have few nursing staff and limited personnel to assist occupants in an evacuation during this time. The fire safety equipment was installed in public areas in the four cases according to the law and was in normal operation according to the subsequent investigations. As shown in Table 3, a number of defense strategies were discussed, as described below.

a.    *Fire location and cause:*

The investigations revealed the fires broke out on different floors. In the arson cases, the fires could not be immediately put out at first, and the second layer of protection failed. In the cases with no casualties, the third layer of protection worked by confining the fires to the rooms where they started. In the electrical incidents, in addition to the automatic sprinkler systems used in the second layer of protection, the third layer of protection worked.

The fires started on the second floor in Case 1 and on the third floor in Case 4, but the former caused casualties, indicating the height of the floor where the fire started was not necessarily the main cause. Among the four cases, two were in the storerooms and the other two were in the wards; two were caused by arson and the other two were caused by electrical fires. However, the results were different. When dividing the cases by the causes of fires, each category has one case that was a serious incident and the other that was an incident with no casualties. The causes clearly affected the results.

b.    *Burning time:*

Table 3 indicates that when the fire could not be distinguished within 15 min, the number of subsequently dispatched fire trucks and firefighters was considerable. In the two serious cases (Cases 1 and 3), it took about an hour from the start of the fires until they were put out. In the two cases with no casualties (Cases 2 and 4), the burning time was less than 15 min. There were no automatic sprinkler systems in the rooms where fires started in two cases (Cases 1 and 2) but there were in the other two cases (Case 3 and 4). The results were different.

Firefighters arrived at the scenes immediately after the fires broke out. More than ten fire trucks and more than 25 firefighters were dispatched for each case, indicating that the fire control unit attached great importance to nighttime fires. As soon as a report was received, massive relief resources were mobilized immediately. In Cases 1 and 2, the emergency vehicles involved were, respectively, 101 (53 fire trucks and 48 ambulances) and 76, and the number of firefighters involved was, respectively, 150 and 236. This illustrates that if the three layers of protection in the fire protection defense-in-depth strategy were breached, the on-site personnel would be mainly used for rescue, although the fire could be put out in about an hour. Despite the firefighters' efforts, many occupants in the institutions unfortunately suffered smoke inhalation or were even killed.

c.    *Fire compartment:*

Mobility-impaired occupants are vulnerable to heat and smoke when they cannot receive assistance. There is usually few nursing staff on duty when nighttime incidents occur. In Taiwan, nighttime personnel are determined based on minimum service requirements rather than from the perspective of fire safety. This is because mobility-impaired occupants in institutions can only move smoothly under the assistance of others.

In the two cases with serious casualties (Cases 1 and 3), the walls of the rooms where the fires started were only connected to the ceilings and were not connected to the floors above, as shown in Figure 3. This was allowed by the fire code when the buildings were constructed. Such buildings would not be a problem for occupants with good mobility, as they could leave the scene quickly after a fire broke out. However, this would be a problem for mobility-impaired or long-term bedridden occupants.

It was clear that housing needs and simplified construction rather than fire safety were considered when the building was built. If walls are not connected to the floors above, heat

and smoke will flow through the space above the ceilings and spread to other rooms. In the two cases with no casualties (Cases 2 and 4), the walls in the rooms where the fires started were concrete and completely connected to the floors above. Therefore, as shown in Figure 4a,b, they were effective as the third layer of protection.

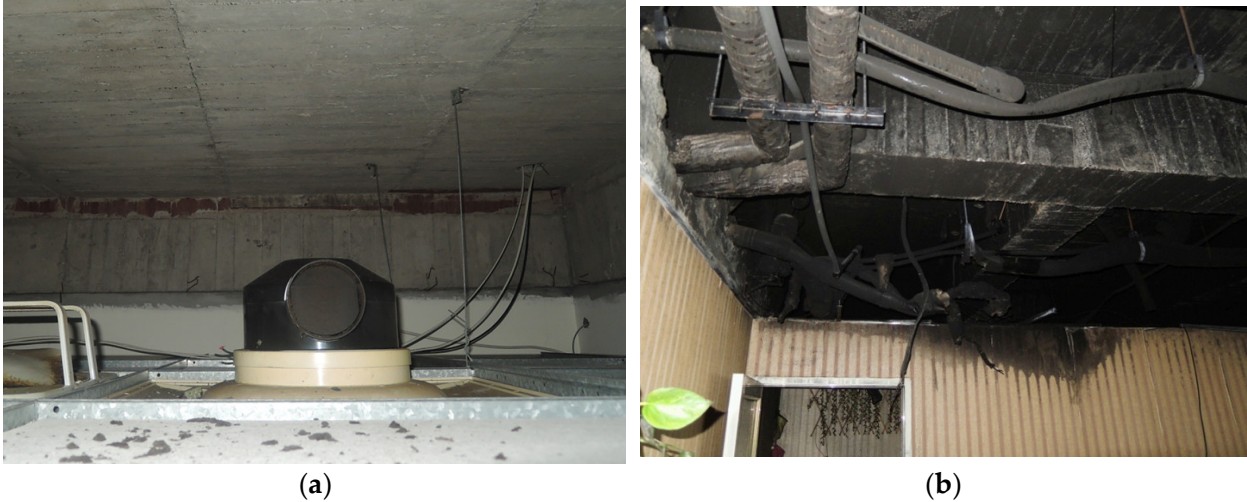

(**a**)  (**b**)

**Figure 4.** (**a**) The wall is connected to the floor structure above (a real photo of the scene, but not an institution mentioned in this paper). (**b**) The wall is not connected to the floor structure above (a real photo of the scene, but not an institution mentioned in this paper).

*d.    Evacuation time ($t_t$) analysis:*

The nursing staff at night are mostly female, and it can be difficult for them to move occupants due to their smaller strength. While waiting, these occupants will be exposed to heat and smoke. Therefore, the fire control unit has to dispatch numerous emergency vehicles and firefighters after receiving a report. If a fire fails to be put out within ten minutes, more relief resources will need to be devoted to reducing casualties.

The probability of disadvantaged groups escaping from a fire is correlated with their walking time during the evacuation. Saburo Horiuchi proposed the analysis of walking ability in institutions, as shown in Table 4 [44]. When a fire breaks out, if the second layer of protection fails to suppress the fire effectively, the fire will expand. At this point, if the third layer of protection fails to control the fire, the occupants of other rooms will require external assistance.

In the initial stage of a fire, if the second layer of protection works or the third layer of protection can suppress the fire, the harm to occupants in institutions will be greatly reduced. If a fire can be put out within 15 min, the need for subsequent support will be greatly reduced. After studying the cases which have successfully responded to the incidents and caused no casualties, this study has come to the above conclusion which has reference and learning values. Unfortunately, this study found that people pay less attention to cases with successful outcomes, while incidents with serious casualties attract widespread attention and are explored on a large scale.

## 5. Conclusions

For countries that have entered an aging society, providing high-quality care services and fire safety in long-term care institutions are important issues. In Taiwan, fire incidents over the years have shown that nighttime fires in care institutions often cause serious casualties, since most occupants in these institutions are mobility-impaired and asleep, and few staff are available to assist in the evacuation. Arson is the most adverse but probable cause for fire accidents in these institutions, and verification and response planning for fire disasters have to be made for disaster prevention and relief. Electrical fires should also be carefully prevented.

People often pay attention to major incidents and actively try to understand their causes and improvement measures; however, it is necessary to learn from nighttime fire incidents with successful outcomes that have no casualties. In this study, the top two serious nighttime fire accidents in long-term care institutions in the past two decades in Taiwan were analyzed using the theory of fire protection defense-in-depth theory and found the fires all broke out between midnight and 6:00 a.m. For comparison, two other nighttime fire cases with similar scenarios but no casualties were also analyzed in depth about the cause of no casualties. The important information collected included the fire location, cause, burning time, rescue personnel, and the number of casualties. The buildings were equipped with fire protection equipment in their public areas according to the fire laws. Two incidents resulted in serious casualties and the other two resulted in relatively light casualties. In the serious incidents, one fire was caused by arson and the other by an electrical fire, with one in a storeroom and the other one in a ward. The situation of the cases with no casualties was the same as that of the serious incidents, but the final results were very different. This study found that the burning time was about an hour in the cases with serious casualties, and that the fires were put out within 15 min in the cases with no casualties.

Institutions where fires occurred have set up hardware equipment and formulated management measures in accordance with laws and regulations. Nursing staff have also undergone regular training in accordance with regulation. The results showed that it is difficult to evacuate occupants after a fire breaks out. When the hazard time ($t_h$) is prolonged due to failure to control or distinguish the fire, by using the timed egress analysis, it was found that a long evacuation time ($t_t$) can result in serious casualties. A well-constructed second layer of defense measures can effectively contain a fire, and a good third layer of protection can prevent fire expansion and casualties. The results could provide a reference for institutions to develop disaster prevention and response measures.

Since there are fewer caregivers at night, it is necessary to notify the fire department as soon as possible to come to the rescue. Therefore, it is necessary to install an active notification system. For long-term care institutions, the use of fire-resistant materials is necessary to avoid the spread of fire. Fire alarm equipment and automatic sprinkler equipment can suppress the growth of fire. If the fire cannot be extinguished or is discovered too late at night, blocking the fire and waiting for firefighters to extinguish it is the last line of defense.

**Author Contributions:** Conceptualization, methodology, validation, investigation, L.-S.W. and R.-N.P.; validation, supervision, resources, writing—original draft preparation, S.-C.W.; project administration, resources, writing—review and editing, C.-H.S.; formal analysis, data curation, W.-C.W. All authors have read and agreed to the published version of the manuscript.

**Funding:** This research was funded by the Ministry of Science and Technology under Grant no. MOST 109-2625-M-992-002.

**Institutional Review Board Statement:** Not applicable.

**Informed Consent Statement:** Not applicable.

**Data Availability Statement:** The data are available upon request from the corresponding author.

**Acknowledgments:** The funding from the Ministry of Science and Technology is acknowledged. We also thank Li-Peng Chen and Jhih-Ang Yang for their help in recording and photographing the scenes.

**Conflicts of Interest:** The authors declare no conflict of interest.

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
