# Peer review of "Characteristic Analysis of Four Major Nighttime Fire Cases on Fire Safety of Long-Term Care Institutions Using Fire Protection Defense-In-Depth Strategy"

_fire, doi:10.3390/fire6030118_

Round 1

Reviewer 1 Report

The article presents the study of different cases of nighttime fire in the form of Case Reports on care institutions for the protection of elderly people. Fires are caused essentially by electrical causes or by arson or criminal hands. The manuscript focuses on the need for people evacuation who have many associated problems and the evacuation times as important factors to be attended. The manuscript is well-written and well-documented with a good number of references. It is a document in the form of highlighted cases that can support case studies with the same characteristics. In my opinion, the article may be published in the present form.

Author Response

First of all, thank you for reviewing our paper and your comments. Please see the attachment in pdf format.

Reviewer 2 Report

O estudo apresenta um tema de grande relevância, a segurança contra incêndio é muito importante e as normas técnicas são desenvolvidas após a ocorrência de desastres. No entanto, faço algumas anotações a fim de contribuir com o trabalho, são elas:

1# O resumo não deixa claro qual método e parâmetros foram usados ​​para comparar as variáveis. Também não apresenta de forma concreta os resultados encontrados no estudo;

2# O documento contém erros graves de formatação, por exemplo, há dois capítulos 1 no texto. Sugiro uma revisão minuciosa e árdua;

3# O capítulo de método é confuso, não apresenta diretamente como foi estruturada a metodologia para a realização do estudo. Por exemplo, o primeiro parágrafo do capítulo que deve apresentar a metodologia traz informações de referência bibliográfica. Os autores devem apresentar a metodologia de forma clara e direta;

4# O capítulo de resultados não apresenta de forma clara os resultados encontrados no estudo. É muito genérico, não fica claro se os autores estão apenas fazendo uma revisão bibliográfica sobre a importância da segurança contra incêndio, ou se foi feito um levantamento quantitativo dos números dos principais desastres ocorridos no país. Os autores precisam revisar completamente este capítulo.

5# O manuscrito traz algumas referências de sites, muitas vezes não se tem muita confiança nesse tipo de citação.

É necessária uma revisão completa da formatação do documento e reformulação do capítulo. Os autores devem procurar apresentar corretamente as partes de um artigo científico. O tema é relevante, o assunto é relevante e tem potencial, porém é necessário adequar o manuscrito.

Author Response

(The authors gave the same response as above.)

Reviewer 3 Report

Article can be accepted after revision according comments:

1. The authors formulate basic hypotheses with the accepted assumptions. It is important to determine the limits of applicability of these assumptions.

2. A similar question arises when applying mathematical expressions.

3. Extensive statistical data are needed for objective conclusions. They must be shown and analyzed in the article.

4. For the objectivity of the conclusions, it is advisable to use a multi-criteria approach, taking into account the totality of the characteristics of the operation of fire systems.

5. The key question is what changes in fire alarm systems are needed for the early identification of fires.

Author Response

(The authors gave the same response as above.)

Reviewer 4 Report

Dear authors, 

Thanks for giving me the opportunity to review the paper. The topic is very interesting and needs a great attention. 

I read twice the paper and I found it interesting and well structured. 

In order to improve the paper I suggest little modifications. In particular :

a) Introduction/literature : well done, but I suggest also more papers: Among others I think 10.15866/iremos.v10i1.11133 could be interesting. 

b) Methods : it's ok, it's very interesting and a practical experience. I like it. However I suggest to indicate in the paper also limits and strongness. 
c) In the conclusions, please indicate better the route.

Author Response

(The authors gave the same response as above.)

Round 2

Reviewer 2 Report

O documento apresenta uma revisão bibliográfica sobre os principais incêndios ocorridos em Taiwan. A análise não traz grande contribuição científica, as informações são totalmente corretas e relevantes, porém, já são informações consolidadas no meio técnico da área de segurança contra incêndio.

Reviewer 3 Report

Article can be accepted